# Lipid Cubic Mesophases Combined with Superparamagnetic Iron Oxide Nanoparticles: A Hybrid Multifunctional Platform with Tunable Magnetic Properties for Nanomedical Applications

**DOI:** 10.3390/ijms22179268

**Published:** 2021-08-27

**Authors:** Lucrezia Caselli, Marco Mendozza, Beatrice Muzzi, Alessandra Toti, Costanza Montis, Tommaso Mello, Lorenzo Di Cesare Mannelli, Carla Ghelardini, Claudio Sangregorio, Debora Berti

**Affiliations:** 1Department of Chemistry, University of Florence, Via della Lastruccia 3, 50019 Sesto Fiorentino, Florence, Italy; lucrezia.caselli@unifi.it (L.C.); marco.mendozza@unifi.it (M.M.); beatrice.muzzi92@gmail.com (B.M.); costanza.montis@unifi.it (C.M.); csangregorio@iccom.cnr.it (C.S.); 2Consorzio Sistemi a Grande Interfase, Department of Chemistry, University of Florence, 50019 Sesto Fiorentino, Florence, Italy; 3Department of Biotechnology, Chemistry and Pharmacy, University of Siena, 1240, I-53100 Siena, Italy; 4ICCOM-CNR, I-50019 Sesto Fiorentino, Florence, Italy; 5INSTM, I-50019 Sesto Fiorentino, Florence, Italy; 6Department of Neuroscience, Psychology, Drug Research and Child Health-Neurofarba-Section of Pharmacology and Toxicology, University of Florence, 50139 Florence, Italy; alessandra.toti@unifi.it (A.T.); lorenzo.mannelli@unifi.it (L.D.C.M.); carla.ghelardini@unifi.it (C.G.); 7Department of Clinical and Experimental Biomedical Sciences “Mario Serio”, Gastroenterology Unit, University of Florence, 50139 Florence, Italy; tommaso.mello@unifi.it

**Keywords:** SPIONs, cubosomes, cubic phases, drug delivery, magnetic properties, nanoparticles, phase transition, lyotropic liquid crystals

## Abstract

Hybrid materials composed of superparamagnetic iron oxide nanoparticles (SPIONs) and lipid self-assemblies possess considerable applicative potential in the biomedical field, specifically, for drug/nutrient delivery. Recently, we showed that SPIONs-doped lipid cubic liquid crystals undergo a cubic-to-hexagonal phase transition under the action of temperature or of an alternating magnetic field (AMF). This transition triggers the release of drugs embedded in the lipid scaffold or in the water channels. In this contribution, we address this phenomenon in depth, to fully elucidate the structural details and optimize the design of hybrid multifunctional carriers for drug delivery. Combining small-angle X-ray scattering (SAXS) with a magnetic characterization, we find that, in bulk lipid cubic phases, the cubic-to-hexagonal transition determines the magnetic response of SPIONs. We then extend the investigation from bulk liquid-crystalline phases to colloidal dispersions, i.e., to lipid/SPIONs nanoparticles with cubic internal structure (“magnetocubosomes”). Through Synchrotron SAXS, we monitor the structural response of magnetocubosomes while exposed to an AMF: the magnetic energy, converted into heat by SPIONs, activates the cubic-to-hexagonal transition, and can thus be used as a remote stimulus to spike drug release “on-demand”. In addition, we show that the AMF-induced phase transition in magnetocubosomes steers the realignment of SPIONs into linear string assemblies and connect this effect with the change in their magnetic properties, observed at the bulk level. Finally, we assess the internalization ability and cytotoxicity of magnetocubosomes in vitro on HT29 adenocarcinoma cancer cells, in order to test the applicability of these smart carriers in drug delivery applications.

## 1. Introduction

Superparamagnetic iron oxide nanoparticles (SPIONs) are responsive to static and alternating magnetic fields and find application in multiple biomedical fields, from biosensing to magnetic resonance imaging (MRI) and targeted drug delivery [1,2,3,4]. Due to their superparamagnetic nature, SPIONs can be guided by an external magnet at the desired target tissue, where they can be conveniently accumulated for in vivo biomedical applications of improved efficacy [5]. Such a selective accumulation minimizes the side effects of SPIONs, while providing enhanced imaging ability for MRI applications and superior therapeutic efficiency for drug delivery [6,7]. In addition, SPIONs convert magnetic energy into heat when exposed to alternating magnetic fields (AMFs), which makes them effective “nano-heaters” for magnetic fluid hyperthermia applications. The hyperthermic efficiency of SPIONs can be maximized by tuning their size and shape [8,9,10], as well as promoting their controlled clustering in supra-particles organizations [11,12,13]. 

The inclusion of SPIONs within a lipid scaffold allows for combining such features with the biocompatibility and versatility of lipid self-assembly [14]. Moreover, their encapsulation into a lipid matrix can enhance their in vitro/in vivo colloidal stability, circulation time and efficiency of internalization into cells [15].

Most of the research so far has focused on the combination of engineered nanoparticles with lipid scaffolds of lamellar nature, such as bulk lamellar mesophases, supported-lipid bilayers and lipid bilayer-enclosed particles [16,17,18]. In the field of nanomedicine, the combination of SPIONs with liposomes is of particular interest, yielding to hybrid nanoparticles called “magnetoliposomes” [19,20,21,22]; such nano-hybrids have been proposed as multifunctional platforms for controlled drug delivery and MRI applications. By comparison, non-lamellar lipid scaffolds, such as lyotropic liquid crystals of cubic symmetry, have received far less attention [23,24,25,26]. The peculiar structure of cubic phases, which features extended hydrophilic and hydrophobic domains, maximizes the encapsulation efficiency of molecules of different polarities, such as glycolipids [27,28], fatty [29,30] and nucleic acids [31,32,33], nutrients and drugs [34]. Importantly, bulk cubic phases can be dispersed into colloidally stable nanoparticles (i.e., “cubosomes”), thanks to Pluronic-based steric stabilizers. Cubosomes preserve the cubic internal structure of the original bulk systems with a lower viscosity, which enables their easy injection for intravenous administration. According to recent studies, cubosomes encapsulate hydrophobic chemotherapeutics such as Paclitaxel [34] and 5-Fluor Uracil [35], with extraordinary high efficiencies, enhancing their bioavailability in biological fluids and reducing their side effects.

The combination of bulk cubic phases and cubosomes with SPIONs has been introduced only recently. In a previous study, we encapsulated hydrophobic SPIONs in 1-monoolein (GMO)/water inverse cubic phases of Pn3m geometry and bulk nature [25]. We found that the presence of SPIONs strongly modifies the thermotropic behavior of neat GMO/water assemblies, shifting the phase boundaries between the cubic Pn3m phase and the inverse hexagonal arrangement (H_II_ phase). The Pn3m-H_II_ phase transition occurs up to 80 °C for neat GMO/water systems and involves a massive release of water (~50%) and hydrophilic molecules possibly contained in the cubic-phase aqueous domains. The presence of SPIONs enables to tune the Pn3m-H_II_ transition temperature, lowering it to physiological values or lower, by embedding proper amounts of nanoparticles. Importantly, we also found that such transition can be promoted at room temperature by applying an AMF, due to the hyperthermic effect of SPIONs.

In this work, we deepen the investigation on this phenomenon, to optimize the potential of GMO/SPIONs/water cubic-phase systems for drug delivery applications. By combining structural and magnetic characterization techniques, we explore the effects of the Pn3m-H_II_ transition on the magnetic properties of SPIONs, embedded into bulk GMO/water scaffolds. Then, we extend the investigation to the colloidal level of magnetocubosomes, i.e., GMO/SPIONs/water nanoparticles of cubic internal structure. Through Synchrotron small-angle X-ray scattering, we explore the possibility of activating the Pn3m-H_II_ transition in magnetocubosomes remotely, i.e., through the application of a low-frequency AMF. Finally, as a preliminary test to evaluate the applicability of magnetocubosomes as smart drug delivery systems, we assess their internalization ability and cytotoxicity in vitro on HT29 adenocarcinoma cancer cells.

## 2. Results and Discussion

### 2.1. Structural and Magnetic Properties of SPIONs

We synthesize hydrophobic SPIONs with a magnetite (Fe_3_O_4_) core and an oleic acid/oleylamine stabilizing shell, according to a well-defined protocol [36,37] (see Section 3.2).

Figure 1 reports a full characterization of SPIONs’ structural features (investigated through SAXS and X-ray Diffraction (XRD) and magnetic response (described by Field Cooled (FC)/Zero Field cooled (ZFC) and field-dependent magnetization curves). The SAXS profile of SPIONs dispersed in hexane (Figure 1a) highlights an average diameter of 3.8 nm, with a 0.3 polydispersity index (see also Appendix A, for TEM images of SPIONs, highlighting a spherical morphology). The XRD pattern (Figure 1b), recorded on a dry SPIONs powder, can be well indexed with the spinel cubic structure of magnetite with space group $Fd\bar{3}m$ (JCPDS No. 75-0449 [38]); the Scherrer analysis of the peak linewidth provides an estimate of the SPIONs’ diameter of 5 nm, (See Appendix A for details), in good agreement with the SAXS analysis, pointing out their single-crystal nature. The magnetic susceptibility (χ) vs. temperature (T) curve, after zero field-cooled (ZFC) and field-cooled (FC) processes (Figure 1c) displays an average blocking temperature T_B_, evaluated as the peak in the ZFC magnetic susceptibility, of 12.5 K. Thus, by the well-known equation derived from the Néel–Brown model for superparamagnetic relaxation, the diameter of the spherical nanoparticles was calculated as 4.3 nm.

Magnetic measurements of nanoparticles (Figure 1d) indicate that SPIONs are superparamagnetic at 293 K and 320 K, meaning that the thermal energy can overcome the anisotropy energy barrier of a single particle, and the net magnetization of the particle assemblies in the absence of an external field is zero. At 2.5 K, below the blocking temperature, an open hysteresis loop can be observed, with a coercivity of 240 Oe and a reduced remanent magnetization of 14 emu/g. Under a large external field, the magnetization of the particles aligns with the field direction and reaches its saturation value, that is 39.2 emu/g at 2.5 K, 31.6 emu/g at 293 K and 30.6 emu/g at 320 K.

### 2.2. SPIONs-Loaded Bulk Cubic Phases

Due to their small core size and hydrophobicity, SPIONs can be easily encapsulated within the hydrophobic portion, i.e., the lipid bilayer, of lipid-lyotropic liquid crystals [39]. Here, we directly embedded SPIONs into GMO/water bulk cubic phases, according to a protocol optimized in our previous studies [23,25,26] (see Section 3.3). We then characterized the thermotropic and magnetic behavior of the resulting hybrids.

#### 2.2.1. Thermotropic Behavior

The thermotropic behavior of GMO/water systems in the absence of SPIONs is well known [25]: in the 25–50 °C range and in water excess, GMO self-assembles in an inverse cubic phase of Pn3m crystallographic space group. This structure has a bicontinuous nature, featuring a single lipid bilayer of negative interfacial curvature, which divides the 3D space into two sets of interwoven aqueous nanochannels. The lattice parameter (d) of such arrangement slightly reduces (~2 nm) with increasing temperature, from 25 °C to 50 °C [25].

In a previous study [25], we observed that the inclusion of hydrophobic SPIONs -of same composition and similar size to the ones employed here-in the GMO/water assembly deeply modifies the thermotropic behavior.

Through SAXS, here we monitored such variations, induced by the inclusion of different amounts of SPIONs (see Appendix A for SAXS profiles recorded at different SPIONs-per-lipid-molecules ratios, ranging from 3.8 × 10^−5^ to 1.3 × 10^−4^) in GMO/water systems.

Figure 2a reports the SAXS profiles of bulk GMO/water systems, assembled with 9.5 × 10^−5^ SPIONs per lipid molecule, as a function of temperature.

At 25 °C, the SAXS profile shows prominent peaks, which are the typical Bragg reflexes of an inverse Pn3m phase; the inclusion of SPIONs produces a reduction in the spacing parameter (d = 9 nm) with respect to the neat GMO/water assembly (d ≅ 10.4 nm) while preserving the cubic arrangement. At 35 °C, the scattering pattern of the Pn3m phase undergoes a low-q shift, associated to a shrinking in the cubic phase’s lattice parameter of 0.7 nm, in agreement with previous reports [25]. Moreover, additional Bragg reflexes emerge, ascribable to an inverted hexagonal phase (H_II_), which consists of cylindrical inverted micelles packed in a hexagonal lattice. The H_II_ phase–Bragg pattern fully replaces the Pn3m one at 40 °C, indicating a complete Pn3m-H_II_ phase transition. Such transition is observed at 80 °C for the case of neat GMO/water systems. In line with previous findings [25], the inclusion of SPIONs into GMO/water systems enables to lower its temperature and finely control it, by tuning the amount of embedded SPIONs. In particular, such temperature can be gradually reduced by increasing the amount of embedded SPIONs (see Appendix A). For the higher amount of SPIONs tested in this work (1.3 × 10^−4^ SPIONs per lipid molecule, see Figure 2b), the Pn3m-H_II_ transition is already complete at room temperature: here, the SAXS profile recorded at 25 °C shows the typical Bragg reflexes of a neat H_II_ phase with a spacing parameter of 4.6 nm. Such arrangement, preserved until 50 °C, undergoes a slight decrease of d (i.e., ~3 nm) by increasing temperature, evident from the higher-q shift of its Bragg peaks’ pattern.

Importantly, the Pn3m-H_II_ phase transition occurs with a massive expulsion of water (~50%) and hydrophilic molecules, possibly dispersed in it; its promotion at physiological temperatures, enabled by the inclusion of SPIONs, has been recently proposed as a new strategy to trigger the release of hydrophilic drugs from cubic phases, for controlled drug delivery applications [25].

The SAXS investigation provides additional information on the arrangement of SPIONs within the lipid scaffold, as a function of temperature. The broad peak in the low-q region of SAXS profiles arises from the spatial correlation of SPIONs within the lipid scaffold and provides the SPION–SPION center-to-center average distance through q_max_ = 2π/d, with q_max_ the value of q at the peak maximum. q_max_ locates at 0.5 nm^−1^ for GMO/SPIONs/water systems of cubic nature, observed in the 25–35 °C range for the 9.5 × 10^−5^ SPIONs/lipid sample (Figure 2a). This corresponds to a SPION–SPION average distance of 12.5 nm, perfectly in agreement with the literature [26]. Remarkably, the SPIONs’ correlation peak undergoes a shift to higher q (i.e., 0.53 nm^−1^ at 50 °C) when raising temperature up to 35 °C, i.e., when the H_II_ phase replaces the Pn3m one. Such structural reorganization brings SPIONs closer to each other, with a new SPION–SPION average distance of 11.8 nm. This value perfectly matches the SPION–SPION distance in the H_II_ phase of the 1.3 × 10^−4^ SPIONs/lipid sample (Figure 2b), which does not vary with temperature. In a recent study [25], we connected such behavior with a “pearl-necklace” like reorganization of SPIONs, which leads to the formation of linear clusters where SPIONs are tightly packed.

#### 2.2.2. Magnetic Behavior

To explore the effects of such realignment on the magnetic behavior of SPIONs, we investigated the magnetic response of GMO/SPIONs/water assemblies as a function of temperature. As control experiment, we measured the susceptibility (χ = M/H) of a neat SPION’s powder as a function of temperature, recorded at 10 Oe (Appendix A). For this control sample, the susceptibility shows a Curie–Weiss decay (χ = C/(T − T_C_) where C is the Curie constant and T_C_ the ordering temperature) into the 20–47 °C range, typical of superparamagnetic nanoparticles [40].

Figure 3a shows the magnetic susceptibility as a function of temperature for the GMO/SPIONs/water assembly at a 9.5 × 10^−5^ SPIONs/lipid, which has a Pn3m structure at 25 °C. The magnetic behavior of this sample significantly deviates from the one of the SPIONs powder: specifically, the magnetization decays according to the Curie–Weiss behavior until 32 °C, over which a clear bump, with a broad maximum located at ca. 34 °C, is observed. Above 40° the susceptibility decay can be described again by a Curie–Weiss law, but with a constant higher by ca. 12%. Interestingly, the temperature range at which the magnetic transition is observed (32–40 °C) perfectly matches the t interval at which the Pn3m-H_II_ phase transition occurs (Figure 2a). The χ vs. T curves showed how the susceptibility changed pre- and post-structural lipidic transition. This effect can be attributed to the new NPs spatial correlation, i.e., pearl-necklace-like. At 32 °C, indeed, the structural rearrangement of the lipid scaffold and the decrease in the lipid bilayer viscosity, with respect to r.t., induces the diffusion of SPIONS within the membrane, which rearrange in the pearl-necklace configuration. This spatial configuration is characterized by stronger and ordered dipolar interactions that lead to the increase of the magnetic susceptibility of the SPIONs. It should be considered that dehydration of the sample is negligible for the time range of the experiments. The same measurement was repeated on the same sample after a thermal cycle (green markers and line Figure 3b). In this case no magnetic transition was observed, and the susceptibility was found to follow the same temperature dependence observed in the first run, above 40 °C, suggesting the process is not reversible and that the hexagonal structure is preserved after cooling to room temperature. This observation is corroborated by a previous report [23], which indicates that the recovery of the Pn3m structure after the transition requires long times (3–4 h).

Figure 3b shows the susceptibility, as a function of temperature, for the GMO/SPIONs/water assembly at 1.3 × 10^−4^ SPIONs per lipid, which has a H_II_ structure in the whole 25–50 °C range. For this sample, the susceptibility curve, as a function of temperature, shows the expected decay without deviation. This result confirms that the hump observed in Figure 3a can be univocally ascribed to the transition of the lipid scaffold, which forces SPIONs to follow the new pearl-necklace arrangement.

To summarize, here we show how the inclusion of SPIONs allows for the tuning of both the structure and magnetic properties of GMO/SPIONs/water systems.

Precisely, the concentration of SPIONs modulates the cubic-to-hexagonal phase transition, which in turns modifies the magnetic response of the hybrids. In addition, this transition represents a possible “structural trigger” to promote the release of hydrophilic drugs contained in the cubic phase. The encapsulation of SPIONs allows for lowering the temperature of such transition to physiological values, enabling its exploitation for in vitro and in vivo applications. Importantly, the magnetic responsiveness of SPIONs also provides the unique opportunity to trigger the Pn3m-H_II_ transition “on-demand” at the biological target; indeed, it has been recently proven [25] that the heat released by SPIONs under the exposure to alternating magnetic fields locally triggers such rearrangement in cubic scaffolds of bulk nature. 

However, the high viscosity of bulk cubic assemblies prevents their direct injection for intravenous administration. “Cubosomes”—e.g., colloidally stable nanoparticles of cubic internal structure—represent the best alternative for drug delivery, preserving the peculiar cubic arrangement with reduced viscosity. 

In the following, we will extend the investigation on GMO/SPIONs/water assemblies to the colloidal level, i.e., to SPIONs-loaded cubosomes (“magnetocubosomes”); we will characterize the thermotropic behavior of such hybrid nanoparticles, comparing it to the one of their corresponding assemblies of bulk nature. Moreover, we will explore their structural response to oscillating magnetic fields, as possible triggers of the Pn3m-H_II_ transition.

### 2.3. SPIONs-Loaded Cubosomes 

#### 2.3.1. Thermotropic Behavior

We prepared colloidally stable cubosomes and magnetocubosomes (at a 9.5 × 10^−5^ SPIONs/lipid concentration) of 200–250 nm in diameter [25], by dispersing bulk hybrid cubic phases in water, in the presence of Pluronic F-127 (Section 3.3). The hydrophilic poly-ethylene oxide blocks of Pluronic stabilize particles through steric repulsions, while the hydrophobic poly-propylene oxide blocks are responsible for the anchoring of the stabilizer to the lipid membrane.

We investigated the internal structure of such particles as a function of temperature, and the amount of embedded SPIONs, extending a previous investigation on similar systems. Measurements were performed through Synchrotron small-angle X-ray scattering, at the ID02 beamline of ESRF Synchrotron Radiation Source, Grenoble (France).

In the 25–50 °C range, cubosomes (Figure 4a) show a primitive cubic internal structure (Im3m phase), in agreement with the literature [41]. Similarly to the Pn3m phase (observed at the bulk level), this arrangement has a cubic bicontinuous nature, but a higher hydration level.

In analogy to bulk systems, the insertion of SPIONs within cubosomes (at 9.5 × 10^−5^ SPIONs per lipid) strongly modifies the thermotropic behavior (Figure 4b). The SAXS profile of hybrid nanoparticles at 25 °C features the presence of a SPIONs-SPIONs correlation peak at low-q (in analogy to bulk hybrid systems) and a pattern of Bragg peaks at intermediate and high q. Such pattern is typical of an inverse cubic Pn3m phase with a lattice parameter of 8.6 nm. However, the presence of a weak additional peak, i.e., the first Bragg reflex of the H_II_ phase, indicates a coexistence od such phase with a less abundant H_II_ structure, already at 25 °C. In line with previous literature [25], this represents the only difference with respect to the bulk system assembled with the same amount of SPIONs. Hybrid particles preserve their cubic structure at higher temperatures like bulk hybrids, and the Pn3m-H_II_ transition is complete only at 40 °C.

Figure 4c reports the effect of increasing the amount of SPIONs to 1.3 × 10^−4^ SPIONs per lipid. Here, the SAXS profile recorded at 25 °C shows the clear Bragg fingerprint of a neat H_II_ phase (i.e., devoid of Pn3m-related signal contaminations). Thus, in perfect analogy to bulk hybrids, increasing the concentration of SPIONs further lowers the Pn3m-H_II_ phase-transition temperature, until its completion at r.t. The H_II_ structure is preserved in the whole 25–50 °C range, with a progressive decrease of the lattice parameter from 5.1 to 4.4 nm.

We can conclude that the thermotropic behavior of GMO/SPIONs/water bulk assemblies is essentially retained at the colloidal level: here, increasing the temperature triggers a magnetocubosomes-to-“magnetohexosomes” phase transition, whose temperature can be tuned in the same way, i.e., by varying the amount of embedded SPIONs.

The Pn3m-H_II_ transition modifies the magnetic properties of GMO/SPIONs/water hybrids, which enables to tune the magnetic response of such systems (see Section 2.2.2). In addition, it can be exploited in drug delivery to burst the release of hydrophilic drugs from the water domains of the cubic phase. In the following, we will explore the possibility to activate such phase transition through a magnetic stimulus, i.e., an alternating magnetic field. Due to their magnetic nature, SPIONs can collect magnetic energy and turn it into heat; local temperature variations induced by SPIONs could possibly elicit a structural response in the thermo-responsive lipid scaffold, in which they are embedded.

#### 2.3.2. Structural Response to Alternating Magnetic Fields

We investigated the structural response of GMO/SPIONs/water nanoparticles to alternating magnetic fields (AMFs) through Synchrotron SAXS (ID02 beamline, ESRF, Grenoble, France). Exploiting a dedicated experimental set-up (See Appendix A) that we previously optimized for measurements on bulk GMO/SPIONs/water assemblies, we monitored possible AMF-induced structural variations “live”, i.e., during the application of the magnetic field. To this purpose, we selected an alternating field of low-frequency (4.55 kHz) and ca. 223 kA/m intensity, which represent safe working conditions for the in vivo application of AMFs [42,43]. Figure 5a collects the SAXS profiles of cubosomes (without SPIONs) at 25 °C and different times of exposure (from 0 to 300 s) to the AFM, recorded as a control experiment. In the presence of the AMF, the Im3m phase of cubosomes is fully preserved, with no major variations detectable in the scattering pattern (see inset of Figure 5a). The AMF application only induces an overall minor shift to higher q (~0.02 nm^−1^) of the SAXS profile, responsible for a slight reduction of the lattice parameter, i.e., from 13.1 nm (in the absence of the AMF) to 12.8 nm (after 300 s of AMF exposure). This is due to a mild Joule effect of the coil, which causes a ≤3 °C increase in temperature within the sample.

Figure 5b shows SAXS profiles of magnetocubosomes at 9.5 × 10^−5^ SPIONs per lipid in the very same conditions (see Appendix A for SAXS profiles of lipid nanoparticles at 1.3 × 10^−4^ SPIONs per GMO molecule). In the presence of SPIONs, the effect of the AMF is dramatically different: the Pn3m phase of magnetocubosomes undergoes a significant reduction in d (~1 nm) only after 90 s of AMF application, evident from the shift to higher q of its Bragg reflections. This shift progressively increases with exposure time, leading to a further reduction in d (~3 nm after 120 s). The shrinkage of the Pn3m structure parallels an intensity increase in the first Bragg reflection of the H_II_, which is barely detectable in the absence of the AMF. This peak reaches an intensity’s maximum at 120 s, concurrently with a strong reduction of the Pn3m-related signal intensity. A complete Pn3m-H_II_ phase transition occurs at 150 s, with the full disappearance of the Pn3m phase fingerprint. 

This dramatic effect can be attributed to the presence of SPIONs in the lipid structure: under AMF exposure; SPIONs behave like “nano-heaters”, i.e., collect magnetic energy and release it in the environment, locally raising the temperature of their lipid scaffold.

The structural variation induced after 150 s of AMF application is comparable to that promoted by raising temperatures to 43 °C, which demonstrates that AMF is as effective as temperature in activating Pn3m-H_II._ In addition, AMF represents a fast trigger of such transition, and offers the unique opportunity of a remote control of drug release at the biological target.

Importantly, the extended q range of the Synchrotron source unveils additional information, contained in the low-q region of SAXS profiles (inset of Figure 5b). Here, the main SAXS feature is the SPION-SPION correlation peak, already observed for the corresponding hybrids of bulk nature. In the absence of AMF, the q position of such peak indicates a mean SPIONs-SPIONs distance of 9.3 nm, which is slightly shorter than in bulk GMO/SPIONs/water systems. The shape and position of the peak are preserved until 150 s of AMF application, i.e., as long as the Pn3m phase is present in the system. The full replacement of the cubic structure with the hexagonal phase parallels an abrupt variation of the scattering signal: a new scattering feature emerges, consisting of a distinct and relatively extended q^−1^ scalar law, beyond the Guinier region of SPIONs, which causes a partial smearing of the SPIONs–SPIONs correlation peak. This feature has been previously observed for magnetocubosomes, under the effect of heating, and connected to their re-organization into linear clusters along the interstitial region between the different cylindrical micelles of the H_II_ array [25]. Such pearl-necklace-like re-organization of SPIONs is templated by the Pn3m-H_II_ transition and aims at minimizing the high frustration energy associated with the packing of lipids into the new H_II_ arrangement [25].

Here, for the first time, we show that the application of an AMF steers the very same reorganization of SPIONs at room temperature. 

Importantly, our studies on bulk GMO/SPIONs/water systems (Section 2.2.2) highlight that such SPIONs’ clustering involves profound modifications to their magnetic response.

Controlling the clustering of magnetic particles has been recently proposed to enhance their heating power [9] and modify their magnetic properties for the desired purpose [44].

Here we demonstrate that a controlled clustering of SPIONs into linear chains is easily switched-on “on-demand”, by exploiting the structural responsiveness of a lipid scaffold; this offers the unique opportunity to finely modulate the magnetic properties of hybrid materials through external stimuli (e.g., temperature and alternating magnetic fields), which is of interest for application in multiple biomedical fields.

### 2.4. SPIONs-Loaded Cubosomes: In Vitro Internalization and Cytotoxicity

Here for the first time, we tested the characteristics of magnetocubosomes in vitro, to assess their potential as hybrid platforms for biomedical applications.

HT29 adenocarcinoma colorectal cells were challenged with such nanoparticles to test their internalization into cells, evaluating the nanoparticles’ capability to easily pass the cell plasma membrane and diffuse into the cytoplasm—a necessary feature to be considered an efficient drug delivery system. Thus, the internalization time of cubosomes (in the absence of SPIONs) was systematically compared to those of hybrid particles containing 9.5 × 10^−5^ SPIONs per lipid (i.e., magnetocubosomes at r.t.) and 1.3 × 10^−4^ SPIONs per lipid (magnetohexosomes at r.t.).

The effect of the coil causes a ≤3 °C increase in temperature within the sample. To determine the time required for the internalization of nanoparticles into cells, 1 × 10^4^ HT29 cells were incubated in suspension with Octadecyl-Rhodamine-B-conjugated nanoparticles (0.01% mol with respect to the GMO amount) at 37 °C. The study was carried out at three different concentrations of nanoparticles (0.6 µg/mL, 6 µg/mL and 60 µg/mL) at 37 °C. Their entrance into cells was evaluated by live-cell microscopy, detecting the fluorescence intensity of the probe encapsulated into the cells after different interaction times with nanoparticles (2, 20 and 40 min). The best time was estimated to be 40 min at the concentration of 60 µg/mL and the images are reported in Figure 6a.

The advantage of cubosomes with respect to “free” hydrophobic drugs can be related to the low bioavailability in biological fluids of non-conjugated therapeutics; in fact, the huge hydrophobic domain allows for the efficient transport of large amounts of drugs (compared to the solubility of the active molecules in bio-fluids) in a relatively small interaction time. Moreover, it is noted in the literature that Pluronic F127, the stabilizing agent of cubosomes, presents a typical enhanced-permeability-and-retention effect (EPR), [45,46,47] due to the PEO blocks of co-polymer, allowing a localization of dispersed liquid crystals in tumor tissues.

Moreover, the biocompatibility of cubosomes, magnetocubosomes and magnetohexosomes was evaluated in terms of observed cytotoxic effect. As largely reported in the literature, the HT29 cell line is commonly used for this type of assessment [48,49,50,51]. HT29 cells (2 × 10^5)^ were incubated in suspension with nanoparticles at different concentrations (0.6 µg/mL, 6 µg/mL, and 60 µg/mL), for 40 min at 37 °C. After incubation, the same volume of each suspension (corresponding to 2 × 10^4^ cells of control) was seeded in a cell plate for 48 h; cell viability was evaluated by MTT assay (Figure 6b). Results suggest that these colloids can be internalized and that they are not toxic for cells with these experimental conditions.

The treatment of cancer cells with nanoparticles exposed to AMF is a strategy that is gaining increasing interest [52,53], even if frequency, time, nanoparticles size and dose can strongly influence the response to AMF [54].

Since, as previously demonstrated, the exposure of NPs-cubosomes and NPs-hexosomes to an AMF should cause an increase of their temperature, HT29 cells have been treated at the higher dose (60 µg/mL) of cubosomes, magnetocubosomes and magnetohexosomes for 40 min and then they were exposed for 60 min to the AMF.

The viability of HT29 cells, plated in MW96 as previously described (based on a control sample without AMF), was assayed after 48 h to evaluate if AMF could cause the death of treated cells. The graph (Figure 6c) does not show a toxic effect in samples under AMF compared with samples without AMF. These results demonstrate the complete biocompatibility of cubosomes, magnetocubosomes and magnetohexosomes with their use on human cells in the presence of AMF.

## 3. Materials and Methods

### 3.1. Materials

Fe(III)-acetylacetonate (97%), 1,2-hexadecanediol (90%), oleylamine (70%), oleic acid (90%), diphenyl ether (99%), denatured ethanol and hexane mixture of isomers employed for the synthesis of hydrophobic SPIONs and 1-oleyl-rac-glycerol (>99.9%) were purchased from Sigma Aldrich (St. Louis, Mo, USA).

### 3.2. Synthesis of Magnetic Nanoparticles

Iron oxide nanoparticles were synthesized according to the methods used by Wang et al. [55] Briefly, 0.71 g Fe(acac)_3_ (2 mmol) were dissolved in 20 mL of phenyl ether with 6 mmol of oleic acid (2 mL) and 4 mmol of oleylamine (2 mL) under N_2_ atmosphere and vigorous stirring. Then, 1,2-hexadecanediol (2.58 g, 10 mmol) was added to the solution. The solution was heated to 210 °C, refluxed for 2 h and then cooled to RT. Ethanol was added to the solution and the precipitate was collected, washed with ethanol and dispersed again in 20 mL of hexane. SPIONs were stored in a dark flask with N_2_ gas on top to prevent oxidation.

### 3.3. Preparation of Bulk and Dispersed Cubic Phases 

Bulk cubic phases in the absence or in the presence of magnetic nanoparticles were prepared as follows: 30 mg of 1-monoolein (GMO) was weighted in a 2mlglass flask with or without the appropriate volume of SPIONs dispersion, to have a concentration of SPIONs per GMO molecule ranging from 3.8 × 10^−5^ to 1.3 × 10^−4^ SPIONs per lipid molecule. About 0.5 mL of hexane was added to dissolve lipids and SPIONs, then the mixture was dried through gentle nitrogen flux, removing the solvent. Lipid or mixture GMO/SPIONs was left under vacuum overnight sheltered by a light source. Lipid film was then hydrate with 50 μL of Milli-Q water and sample was then centrifuged alternating cycles with cap facing upward or downward. Bulk systems were left in a dark place for at least 12 h in order to stabilize the system. Cubosomes and SPIONs-loaded cubosomes were prepared following the procedure of bulk mesophase preparation until the film dried under vacuum. Then, 8 mg of Pluronic F-127 was added to the dry films and the mixture was heated in a water bath at 70 °C for 5′ to melt the Pluronic F-127 and then vortexed for 5′. Five cycles of heating-vortexing were carried out and then 500 µL of H_2_O preheated at 70 °C was added. The dispersion was then sonicated in a bath-sonicator at 59 kHz and 100% of power for 6 h, to homogenize the system.

### 3.4. Small-Angle X-ray Scattering

SAXS measurements were carried out on a S3-MICRO SAXS/WAXS instrument (HECUS GmbH, Graz, Austria) which consists of a GeniX microfocus X-ray Sealed Cu Ka source (Xenocs, Grenoble, France) power 50 W providing a detector focused X-ray beam with k = 0.1542 nm Cu Ka line. The instrument is equipped with two one-dimensional (1D) position sensitive detectors, (HECUS 1D-PSD-50 M system) each detector is 50 mm long (spatial resolution 54 lm/channel, 1024 channels) and covers the SAXS q-range (0.003 < q < 0.6 Å^−1^) and the WAXS q-range (1.2 < q < 1.9 Å^−1^). The temperature was controlled by means of a Peltier TCCS-3 Hecus. SAXS curves of bulk cubic phase were recorded at 25, 30, 35, 40, 45 and 50 °C in a solid-sample holder. Dispersion of SPIONs were recorded in a glass capillary.

Synchrotron small angle X-ray scattering experiments were carried out at the beamline ID02 at the European Synchrotron Radiation Facility (ESRF, The European Synchrotron, 71 Avenue des Martyrs, CS40220, 38043 Grenoble Cedex 9,) [56]. A scattering vector (of magnitude q) range of 0.007≤ q ≤0.2 nm^−1^ was covered with two sample–detector distances (1 and 10 m) and a single-beam setting for an X-ray monochromatic radiation wavelength λ = 0.10 nm (12.46 keV). The beam diameter was adjusted to 72.4 μm in the horizontal (x) direction and 42.3 μm in the vertical (y) direction (full width at half-maximum at the sample). Assuming a Gaussian distribution, the portion of the beam hitting outside the channel can be estimated. When the centered channel is but ∼0.3% closer to the edge, more beam overlaps it. The beamstop diameter was 2 mm. As a detector, a 2D Rayonix MX-170HS with a pixel size of 44 × 44 μm^2^ was used, which was housed in an evacuated flight tube, at a sample-to-detector distance of alternatively 10 m (leading to an available q-range of 0.007–0.02 nm^−1^) or 1 m (leading to an available q-range of 0.07–0.2 nm^−1^). The exposure times for the background and sample measurements were 0.5 s, for the case of 1 m sample-to-detector distance, and 0.3 s, for the case of 10 m sample-to-detector distance. Measured scattering patterns were normalized to an absolute intensity scale after applying standard detector corrections and then azimuthally averaged to obtain the one-dimensional intensity profiles, denoted by I (q).

SAXS profiles were collected at different temperatures in the 25–49 °C range, with 2 °C steps from one profile to the next. Equilibration time at each temperature was 5 min. In-situ structural detection upon AMF (amplitude: ca. 223 kA/m, frequency: 6.22 Hz) was performed as in the setup showed in Appendix A.

### 3.5. X-ray Diffractometer

The structure of the NPs was investigated by X-ray powder diffraction (XRD) using a Bruker New D8 ADVANCE ECO (Bruker, Billerica, Massachusetts, USA) diffractometer equipped with a Cu Kα radiation. The measurements were carried out in the range 20°−70°, with a step size of 0.03° and a collection time of 1 s.

### 3.6. Measurement of Magnetic Properties

The magnetic properties of the NPs and bulk phases in the absence or in the presence of SPIONs were measured on a Quantum Design MPMS SQUID magnetometer with 50 kOe maximum field. The magnetization versus temperature measurements were performed in zero-field-cooled (ZFC) and field-cooled (FC) conditions with a 50 Oe probe field. The hysteresis loops were measured at increasing temperatures after FC in 50 kOe from 310 down to 4 K.

### 3.7. Cell Culture

Colorectal adenocarcinoma cancer cells HT29 were purchased from European Collection of Cell Culture (ECACC). Cells were routinely cultured in Dulbecco’s Modified Eagle’s Medium (DMEM)—high glucose (4500 mg/L) supplemented with 2 mM glutamine, with penicillin (100 U/mL) and streptomycin (100 μg/mL), and with 10% fetal bovine serum (FBS, Euroclone). Cells were incubated at 37 °C in a humidified atmosphere of 5% CO_2_- 95% air.

### 3.8. Cubosomes’ Internalization Assay

1 × 10^4^ colorectal adenocarcinoma cells HT29 were plated and 24 h later were treated with culture medium in the presence or absence of different concentrations (0.6 µg/mL, 6 µg/mL, and 60 µg/mL) of octadecyl-rhodamine, conjugated cubosomes loaded with 9.5 × 10^−5^ SPIONs per lipid molecule. Cells were incubated 2, 20 and 40 min at 37 °C in humidified 5% CO_2_ atmosphere and then were washed with PBS 1× and imaged with a Leica AM 6000 microscope equipped with a DFC350FX camera and 40 × 0.60NA air objective. All images were equally adjusted for display purposes using Fiji-Image J smart LUT [57].

### 3.9. Incubation with Cubosomes and SPIONs-Loaded Cubosomes

2 × 10^5^ cells were incubated in suspension with different concentrations (0.6 µg/mL, 6 µg/mL, and 60 µg/mL) of cubosomes, magnetocubosomes and magnetohexosomes, for 40 min at 37 °C and 5% CO_2_ to allow the internalization of the molecules. In order to evaluate their toxicity, the same volume of each suspension (corresponding to 2 × 10^4^ cells of control) was seeded in MW96 in triplicate for 48 h and cell viability was assayed. Otherwise, in order to evaluate the effect of the AMF, cells were exposed to an alternating magnetic field for 30 min after the internalization.

### 3.10. Cell Viability Assay

5 mM MTT (3-(4,5-Dimethylthiazol- 2-yl)-2,5-diphenyltetrazolium bromide (Sigma Aldrich, St. Louis, MO, USA) was added to cells and incubated for 1 h at 37 °C. Cells were suspended in 200 μL of dimethyl sulfoxide: wavelength measuring was performed at 595 nm using a spectrophotometer.

## 4. Conclusions

In summary, we here investigated the structural and magnetic properties of GMO/SPIONs/water systems, with the aim of optimizing such hybrids for biomedical application, especially in the field of drug delivery. We found that a structural reorganization of the bulk lipid moiety—i.e., the Pn3m-H_II_ phase transition—increases the magnetic susceptibility of SPIONs, which is connected to their alignment into linear chains. Such controlled clustering of SPIONs, steered by a lipid scaffold, offers the unique opportunity to tune the magnetic properties of GMO/SPIONs/water systems for the desired biomedical purpose, e.g., enhancing their heating power for magnetic hyperthermia treatments. Importantly, we extended our findings from bulk liquid-crystalline phases to colloidal dispersions, i.e., to lipid/SPIONs nanoparticles with cubic internal structure (magnetocubosomes). Here, we demonstrated that heat released by SPIONs under the application of an AMF triggers a magnetocubosomes-to-magnetohexosomes phase transition, which activates the very same linear clustering of SPIONs. Furthermore, the magnetocubosomes-to-magnetohexosomes phase transition occurs with massive water expulsion, which activates the release of hydrophilic drugs, possibly embedded into cubosomes. Finally, we assess, in vitro, HT29 adenocarcinoma cancer cells’ capability for internalizing magnetocubosomes and the cytotoxicity of such hybrid particles, finding that they are easily internalized by cells and they do not cause cytotoxic effects. Overall, we showed that GMO/SPIONs/water assemblies are endowed with structural and magnetic responsivity to stimuli such as AMFs, which makes them promising smart hybrid materials for biomedical application. Specifically, such features could be exploited for triggering, “on-demand”, the release of drugs loaded into GMO/SPIONs/water nanoparticles, and/or to optimize the magnetic properties of magnetocubosomes for magnetic-fluid hyperthermia applications.

## Figures and Tables

**Figure 1 ijms-22-09268-f001:**
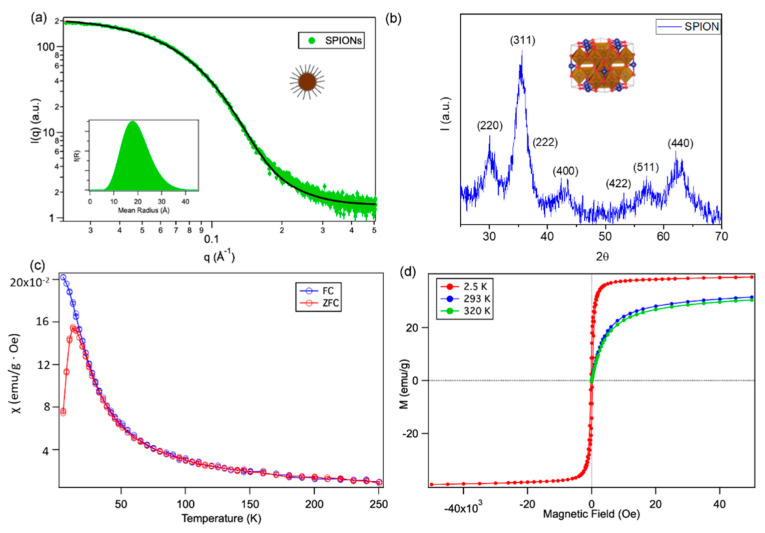
Structural and magnetic characterization of SPIONs: (**a**) SAXS profile of SPIONs dispersed in hexane, together with the best curve-fit according to the SphereSchultz model by NIST (See Appendix A); (**b**) X-ray Diffractogram (XRD) pattern of a dry powder of SPIONs of magnetite *(*Fe_3_O_4_, JCPDS No. 75-0449 [38]); (**c**) zero field-cooled (ZFC) (red circles) and field-cooled (FC) (blue circles) curves, recorded for a dry SPIONs’ powder, at 50 Oe. (**d**) Field-dependent magnetization curves recorded for a dry SPIONs’ powder at 2.5 K (red line and markers), 293 K (violet line and markers) and 320 K (green line and markers).

**Figure 2 ijms-22-09268-f002:**
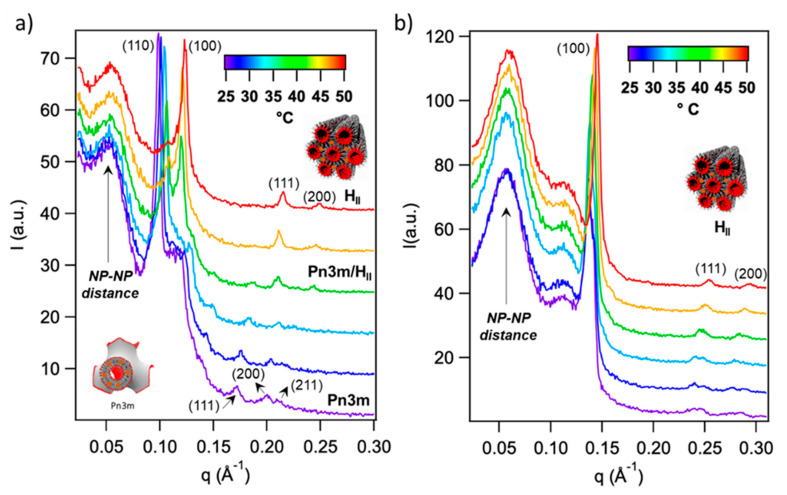
Thermotropic of behavior of GMO/water/SPIONs assemblies. SAXS profiles of bulk GMO/water liquid crystals loaded with (**a**) 9.5 × 10^−5^ and 1.3 × 10^−4^ (**b**) SPIONs per GMO molecule in the temperature range 25–50 °C (see Appendix A for the calculation of the spacing parameters). The Bragg reflexes of cubic and hexagonal phases are indexed in the graph with the corresponding Miller indices. The SPIONs-SPIONs correlation peak at low-q is indicated with a black arrow in the graph. The cubic and the hexagonal arrangements are sketched in the graph.

**Figure 3 ijms-22-09268-f003:**
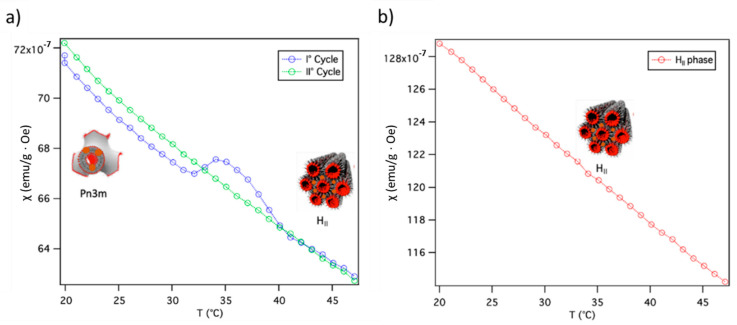
Magnetic susceptibility of GMO/water systems doped with (**a**) 9.5 × 10^−5^ and 1.3 × 10^−4^ (**b**) SPIONs per GMO molecule in the 20–47 °C range, measured on increasing temperature: in subfigure a, magnetic susceptibility data recorded in two subsequent runs, i.e., on the freshly prepared sample (blue), and after it underwent the phase transition (green), are shown. In subfigure b, the magnetic behavior of the hexagonal mesophase is reported, where no phase transition can be detected. The magnetic susceptibility was recorded with a 10 Oe applied field.

**Figure 4 ijms-22-09268-f004:**
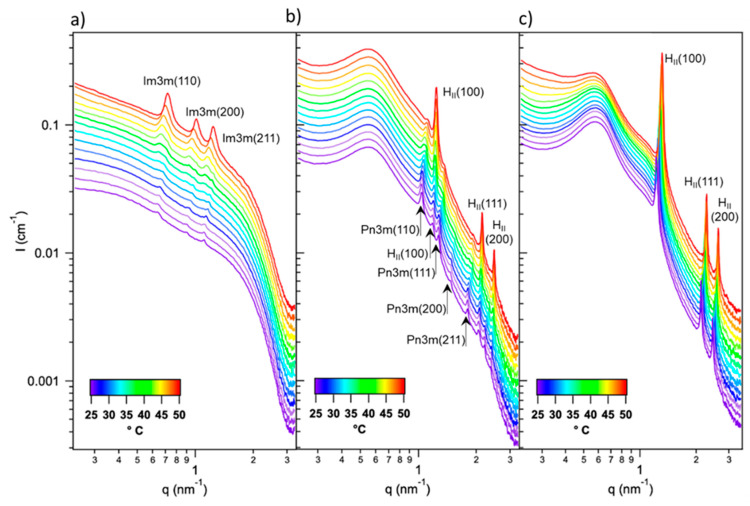
SAXS profiles of lipid nanoparticles doped with (**a**) 0, (**b**) 9.5 × 10^−5^ and (**c**) 1.3 × 10^−4^ SPIONs per GMO molecule, in the 25–50 °C temperature range (see Appendix A for the calculation of the spacing parameters at each temperature). The Bragg reflexes of Im3m, Pn3m and H_II_ phases are indexed in the graph with the corresponding Miller indices. The SPIONs–SPIONs correlation peak is visible as a broad band at low-q in graphs (**b**,**c**).

**Figure 5 ijms-22-09268-f005:**
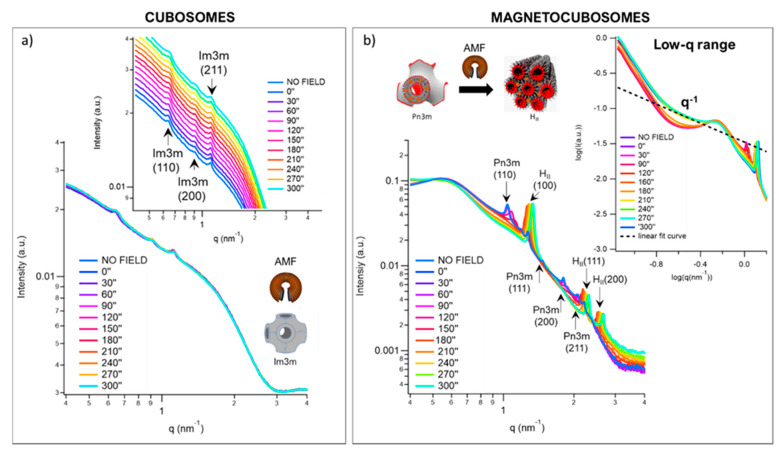
SAXS profiles of cubosomes (**a**) and magnetocubosomes at a 9.5 × 10^−5^ SPIONs per GMO molecule (**b**), recorded at 25 °C under the exposure to the AMF. Each SAXS profile corresponds to a different time of application of AMF. The Bragg reflexes of Im3m (**a**), Pn3m and H_II_ (**b**) phases are indexed in the graphs with the corresponding Miller indices. The inset in (**a**) reports a magnification of the SAXS profile of cubosomes, where the Bragg reflexes of the Im3m phase are highlighted. The inset in (**b**) reports the detail of the low-q region of SAXS profiles of magnetocubosomes, acquired under the exposure to AMF. At 210 s of AMF application, SPIONs self-organize into a pearl-necklace like structure, evident from the appearance of a q^−1^ scalar law (highlighted in the graph) in the scattering profile of hybrid nanoparticles.

**Figure 6 ijms-22-09268-f006:**
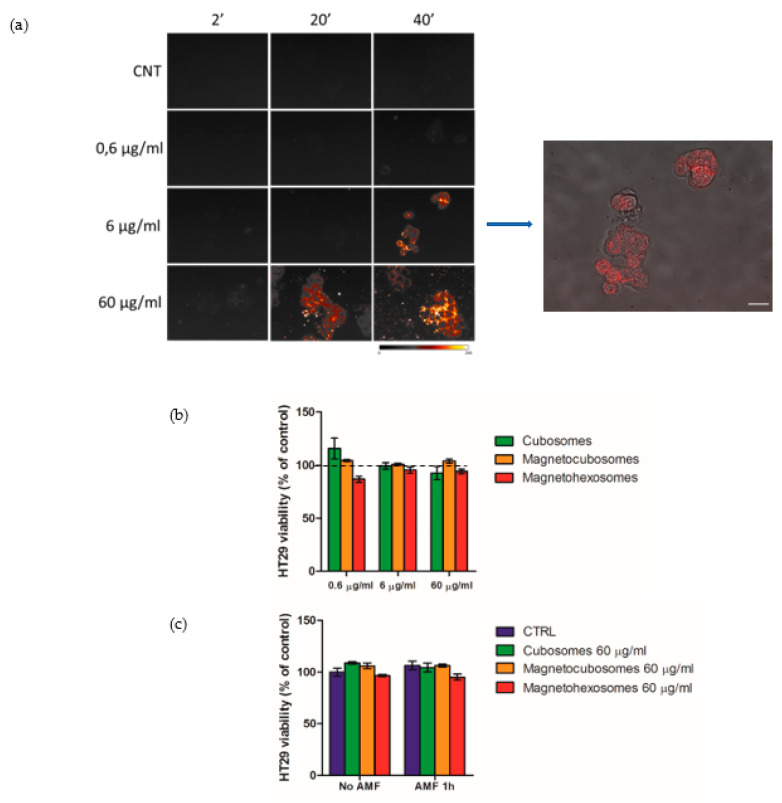
Live-cell fluorescence images of Octadecyl-Rhodamine-B-conjugated cubosomes internalization in HT29 cells at 2, 20 and 40 min at different concentrations (0.6 µg/mL, 6 µg/mL, and 60 µg/mL), the arrow indicates a merge of brightfield and fluorescence images showing cubosome localization in HT29 cells; scale bar = 20 µm. (**a**). Cell viability of HT29 treated with different concentration (0.6 µg/mL, 6 µg/mL, and 60 µg/mL) of cubosomes, magnetocubosomes and magnetohexosomes compared to control set as 100% (see black dotted line in the picture) 48 h after seeding (**b**). Cell viability of HT29 treated with cubosomes, magnetocubosomes and magnetohexosomes (60 µg/mL) under the alternating magnetic field, assayed 48 h after seeding (**c**).

## Data Availability

Data is contained within the article or Appendix A.

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
