# Peer review of "Lipid Cubic Mesophases Combined with Superparamagnetic Iron Oxide Nanoparticles: A Hybrid Multifunctional Platform with Tunable Magnetic Properties for Nanomedical Applications"

_ijms, 2021, doi:10.3390/ijms22179268_

Round 1

Reviewer 1 Report

The authors developed a series of experiments to further investigate the structural and magnetic properties of GMO/SPIONs/water systems that they have established previously. By controlling alignment of lipid moiety into linear chains, they found that GMO/SPIONs/water systems can provide alternative magnetic properties for various treatment desire.

I have one major concern for related conclusion from figure 6. The authors claim that magnetocubosomes are compatible in HT29 adenocarcinoma cancer cells; however without cytotoxicity in healthy cells, it is hard to claim biocompatibility of GMO/SPIONs/water systems. The authors need to provide negative control to support the biocompatibility conclusion.

Reviewer 2 Report

The manuscript by Caselli et al. “Lipid cubic mesophases combined with Superparamagnetic Iron Oxide Nanoparticles: a hybrid multifunctional platform with tunable magnetic properties for nanomedical applications” is interesting and demonstrates the synthesis of hybrid magnetic materials for nanomedical applications”. This manuscript is interesting and requires major revision before its publication as follows:

Comments.

  1. Line 42-43, elaborate this sentence with few examples of “application in multiple biomedical fields”. Also, provide brief information on specific characteristics of nanoparticles, their limitations, advantages, and mechanisms (in vivo) for biotechnological application especially drug delivery. 
  2. Authors may provide the SEM or TEM images to visualize the structural morphology.
  3. Lines 119-126, please provide information about XRD results and discussion along with the addition of JCPDS No. 75-0449 for Fe3O4 particle in the text (in Fig. 1b also) i.e. doi: 10.1021/acsami.6b05165.
  4. Based on the experimental results, the mechanisms of biocompatibility “enter into cell” may be illustrated. 
  5. In addition to biocompatibility (in vitro), the immunogenic (in vivo) properties of materials are also important for biomedical applications. However, these features of materials can be also experimentally proved for effective biomedical application. 
  6. Figure 6a, It is not clear, maybe additional images (large dimension) were provided. Also, the clear internalization of hybrid nanoparticles may be justified. Also, cell viability data of iron oxide (Fe3O4) particles as control may be added.
  7. The discussion may be more polished especially section 2.4 with citations details.
  8. Conclusions may be provided in a single paragraph. Provide (one sentence, only) future perspective by specifying an example of the biomedical area at the end of the conclusion.

Round 2

Reviewer 2 Report

Accept as is